# Phenolic-Compound-Rich *Opuntia littoralis* Ethyl Acetate Extract Relaxes Arthritic Symptoms in Collagen-Induced Mice Model via Bone Morphogenic Markers

**DOI:** 10.3390/nu14245366

**Published:** 2022-12-17

**Authors:** Zainab H. Almansour, Hairul-Islam Mohamed Ibrahim, Rabab S. Hamad, Heba Ibrahim Abd El-Moaty

**Affiliations:** 1Biological Sciences Department, College of Science, King Faisal University, Hofuf 31982, Saudi Arabia; 2Pondicherry Centre for Biological Science and Educational Trust, Pondicherry 605004, India; 3Central Laboratory, Theodor Bilharz Research Institute, Giza 12627, Egypt; 4Medicinal and Aromatic Plants Department, Desert Research Center El-Mataria, Cairo 11753, Egypt

**Keywords:** rheumatoid arthritis, *Opuntia littorallis*, polyphenols, NF-κB, COX-2, miRNAs

## Abstract

Rheumatoid arthritis (RA) is an autoimmune disease that causes inflammation and progressive joint dysfunction. *Opuntia littoralis (OL*) has a high nutritional content and is thought to offer a number of health advantages. We aimed to evaluate the anti-arthritic potential of *OL* extracts against collagen-induced arthritis (CIA). We designed three *OL* cladode fractions from the concentrated aqueous extract: hexane, ethyl acetate (EAE), and hydro alcohol (HAE). We investigated the nitric oxide and MDA levels of EAE against lipopolysaccharide-induced RAW264.7 cells; then, we administered EAE to the mice with CIA to confirm the anti-inflammatory effects against RA. HPLC analysis of the *OL* extracts showed a high concentration of phenolic compounds in EAE. Treatment with EAE (10 and 20 mg/100 g body weight of mice) after 10 days of immunization with collagen showed a significant inhibition of joint inflammation, paw swelling, and edemas. MDA and cytokine levels (IL-1β, IL-6R, IL-6, IL-17, and IL-23) were significantly reduced. EAE effectively ameliorated COX-2, NF-kB, STAT-3, PTEN, and RANKL expression. *OL*-EAE therapy significantly upregulated the expression of miR-28 and miR-199a. In conclusion, the anti-inflammatory actions of *OL*-EAE altered the cellular localization of the inflammatory mediators, therefore preventing joint inflammation via partial epigenetic and metabolic regulations in experimental mice.

## 1. Introduction

Rheumatoid arthritis (RA) is a systemic and chronic inflammatory condition brought on by an autoimmune reaction to a triggering agent in the environment [1]. The pathophysiology of RA includes synovial cell hyperplasia and the endothelial cell activation involved in the progression of cartilage degeneration followed by bone degeneration [2]. The genes involved in the inflammatory response are activated by these inflammatory mediators, and this leads to tissue deterioration [3].

Immune pathogenesis in bone deformation is greatly influenced by cell regulatory markers. Recently, it was discovered that receptor activator of nuclear factor kappa-B ligand (RANKL) promotes osteoclast differentiation [4]. The nuclear factor kappa light chain enhancer of activated B cells (NF-κB) and the signal pathway plays a crucial role in the inflammatory process of RA because it is unusually active in RA synovial tissue. NF-κB regulates a variety of biomolecules, including cyclooxygenase-2 (COX-2) and pro-inflammatory cytokines such as interleukins (IL)-1β and IL-6 and tumor necrosis factor (TNF)-α [5,6,7,8]. Autoimmune arthritis was induced in this model by immunization with an emulsion of complete Freund’s adjuvant and type II collagen (CII).

In addition, macrophages play a crucial role in the pathophysiology of RA and make up the majority of the cell population in the inflamed synovium [9]. Inflammation and articular damage in RA are primarily caused by tissue-degrading enzymes released by macrophages, which are also a primary source of pro-inflammatory cytokines [3,10]. The deterioration of bone and cartilage is correlated with increased synovial macrophage infiltration. These cellular and physiological changes are well observed in collagen-induced arthritis (CIA) mouse models, one of the most commonly studied autoimmune models of RA. The pathophysiological conditions mimic human arthritis conditions and induce them better than adjuvant-induced and carrageenan-induced arthritic models do [11].

miRNAs, which are noncoding short RNAs, participate in a vast network of intricate epigenetic interactions and have a variety of functions in the setting of the development of RA serum levels of many miRNAs and can be used to predict how well RA treatments may work [12]. In experimental arthritic mice, the efficacy of miRNA analogs or antagonists as therapeutic regimens has been proven [13,14].

Disease-modifying antirheumatic drugs (DMARDs) and biologics are currently the main RA therapies [15]. However, only about a third of RA patients have their condition adequately controlled [16]. The anti-inflammatory and painkiller drugs that are available today have alarming side effects and unforeseen consequences [17]. New antirheumatic drugs with high efficacy are needed. Because plant remedies contain a wide range of secondary metabolites with a variety of biological roles, scientists have made efforts to learn more about the enigmatic effect they have on inflammation and pain [18]. Consequently, numerous studies have demonstrated the effectiveness of herbal remedies in treating RA [8,19,20,21].

The *opuntia* species, a member of Cactaceae family, is classified as a folk medicine because it has been used as food and as a therapy for a variety of ailments, in addition to its antibacterial activity [22,23]. *Opuntia* serves as a source for numerous distinct phyto-constituents, including quercetin, myricetin, isorhamnetin, vitexin, orientin, rutin, and kaempferol [23,24].

From the cladode extract of *Opuntia littoralis (O. littoralis)*, it was possible to separate 8-carbomethoxy-5 hydroxyl 6-methylisoflavone, kaempferol, quercetin, gallic acid, ferulic acid, and chlorogenic acid. *O. littoralis* has a high nutritional value, since water makes up the majority of cladodes and fruits, followed by carbohydrates, ash, proteins, and lipids [25]. The herb has also demonstrated antibacterial and diabetic effects [26]. The current study aims to investigate the potential antioxidant and anti-inflammatory effects of *Opuntia littoralis* cladode extracts in regulating collagen-induced arthritis both in vivo and in vitro.

## 2. Materials and Methods

### 2.1. Plant Material

*Opuntia littorallis* (*OL*) cladodes were collected from Al-Ahsa, Saudi Arabia, in March 2021. The plant specimens were authenticated by Prof. Adel Kamel Youssef (Department of Medicinal and Aromatic Plants, Desert Research Center), and a voucher specimen was deposited in our laboratory. The fresh plant cladodes were cleaned, baked in an oven at 50 °C until complete dryness to safeguard sensitive active components [27], and then crushed to a fine powder (80-mesh sieves) and prepared for use in the experiments.

### 2.2. Chemicals and Reagents

Polar and non-polar solvents were purchased from Fischer scientific (Waltham, MA, USA). Biochemical kits and ELISA kits for cytokine estimation were purchased from Biovision technologies and Cayman chemicals (Ann Arbor, MI, USA). PCR reagents were purchased from Invitrogen and Thermo scientific (Waltham, MA, USA). Primers were purchased from Macrogen (Seoul, S. Korea). Primary and secondary antibodies were purchased from Abbexa and Biorybt peptide (Cambridge, UK). Complete Freund’s adjuvant (CFA) was purchased from Sigma Chemical Co. (St. Louis, MO, USA). All other chemicals and reagents, unless specified otherwise, were from Sigma scientific chemicals.

### 2.3. Preparation of OL Cladode Extract

In the next step, 2 kg of dried *OL* cladodes was extracted with 70% aqueous ethanol (3.5 L, three times). The alcoholic extract was then filtered and dried under low pressure [28]. Using a separating funnel device, the concentrated aqueous extract was progressively fractionated using hexane, ethyl acetate, and 70% ethyl alcohol in order to increase polarity. Each fraction was dried out by evaporating it under reduced pressure. The dried extracts were kept in an airtight screw-cap tube at room temperature for further analysis. The dry mass of the concentrated extract served as the basis for all of the tests.

### 2.4. Phytochemical Screening

The preliminary phytochemical screening of the *OL* cladode extracts was conducted to determine the several active components in hexane, ethyl acetate, and 70% hydro alcoholic extract [28]. Dragendorff’s test was performed to determine the presence of alkaloids [29]; Molisch and Biuret tests were employed to determine the presence of carbohydrates [30]. Pew and alcohol sodium hydroxide tests were used to identify flavonoids, and the ferric chloride test was used to determine tannins [29].

#### 2.4.1. Determination of Total Phenolic Acids

The total phenolic acid content was assessed using the Folin–Ciocalteu reagent assay followed by Singleton and Rossi’s approach, as per [31]. At 765 nm, the absorption wavelength was measured. The standard utilized was gallic acid. According to the milligrams of gallic acid equivalent per milligram of extract (μg GAE/mg extract), the phenolic acid content was calculated.

#### 2.4.2. Determinations of Total Flavonoid Content (TFC)

The TFC in the three extracts of *OL* was measured by using the colorimetric method by Zou et al. [32] with minor modifications; rutin was employed as the standard. First, a 1.5 mL EP tube containing 60 μL of a sufficiently diluted sample solution, 360 μL of methanol, and 20 μL of a 5% NaNO_2_ solution was gently shaken for 6 min. After standing for another 6 min, 40 μL of 10% AlCl_3_–6H_2_O was added, followed by the addition of 120 L of 4% NaOH solution. A UV/VIS spectrophotometer was used to measure the reaction mixture’s absorbance at 510 nm after 15 min (UV-1100, MAPADA, Shanghai, China). Methanol was used as the blank. TFC was calculated as milligrams of rutin equivalents per gram of sample (mg RE/g sample) for each sample. This experiment was performed three times on each sample.

#### 2.4.3. Investigation of Polyphenolic Compounds Using HPLC

The ethyl acetate and 70% aqueous ethanolic extracts of the *OL* cladodes were analyzed using high-performance liquid chromatography [33]. HPLC analysis was carried out using an Agilent 1260 series. The separation was carried out using an Eclipse C18 column (4.6 mm × 250 mm i.d., 5 μm). The mobile phase consisted of water (A) and 0.05% trifluoroacetic acid in acetonitrile (B) at a flow rate 0.9 mL/min. The mobile phase was programmed consecutively in the following linear gradient: 0 min (82% A); 0–5 min (80% A); 5–8 min (60% A); 8–12 min (60% A); 12–15 min (82% A); 15–16 min (82% A); and 16–20 (82%A). The multi-wavelength detector was monitored at 280 nm. The injection volume was 5 μL for each of the sample solutions. The column temperature was maintained at 40 °C.

### 2.5. Antioxidant Assays of OL Extract

#### 2.5.1. DPPH Assay

The DPPH assay was carried out according to the methodology of Loganayaki et al. [34], with some modifications for the crude ethanol extracts from three different organs as triplicate values at 517 nm. Methanol was used as a blank. The percentage of inhibition was calculated by standard procedure.

#### 2.5.2. FRAP Assay

The FRAP assay was carried out according to methodology of Yamaguchi et al. [35]. The absorbance was 593 nm, and FeSO_4_·7H_2_O was used as the standard. FRAP activity was expressed as mM Fe^2+^/g of sample.

### 2.6. Anti-Inflammatory Activities of OL Extracts

The synovial-like fibroblast cells were isolated from naïve mice using the miltenyi 105 cell marker isolation kit. Murine macrophage RAW264.7 cells were collected from the American Type Culture Collection (ATCC) and maintained. Recombinant IL-6 was used for the induction of synovial cells, and LPS was used to induce an inflammatory response in the macrophage cells. Nitrite was quantified according to the Griess reagent protocol [36], with the absorbance at 540 nm. Briefly, the cells were transferred into a 96-well plate (4–6 × 10^4^ cells in each well) and incubated for 24 h. Then, those cells were cultured with EAE and HAE extracts of for 12 h. Next, 2 ng/mL of IL-6 and 5 ng/mL of LPS were added to those cells and cultured for another 2 h. Induced cells were lysed using triton X 100 and lysate with supernatant mixed with the Griess reagents (1% sulfanilamide and 0.1% naphthylethylene diaminedihydrochloride in 5% H_3_PO_4_). After 15 min of reaction, the mixture absorbance was read at 540 nm with a microplate reader.

### 2.7. Acute Oral Toxicity Research

The study determining acute oral toxicity was carried out according to ethical guidance protocols. The mice were starved overnight and only given water. *OL*-crude extract dosages of 500 mg/kg of body weight were dissolved in 0.15 mM sodium citrate buffer and administered orally for 3 days. The behavior and expression of the mice were monitored for seven days after the oral treatment.

### 2.8. Preparation of Arthritis Induction Solution

The arthritic induction emulsion was prepared by following 25 μg type II collagen (Molequle On, New Zealand) in 4 mg/mL of complete Freund’s adjuvant (CFA). Collagen was dissolved in 10 mM acetic acid with 4 mg/mL of CFA and was emulsified using a sterile T connector syringe.

### 2.9. Induction of Arthritis in Mouse Model

Swiss albino mice (6–8 weeks old and an average weight of 22 ± 2 g) were maintained at the College of Science, King Faisal University, Saudi Arabia. Animals were assigned into 4 experimental groups of 6 animals per group after acclimatization. The study duration was 29 days. Group I represented the untreated controls and were given oral distilled water. Group II was given collagen 25 µg/25 g of mouse body weight (BW). Group III received collagen 25 µg/25 g of mouse BW with *OL*-EAE extracts as oral gavage (10 mg/100 g) according to mouse BW, and Group IV received collagen 25 µg/25 g of mouse BW with *OL*-EAE extract as oral gavage (20 mg/100 g) according to mouse BW. Briefly, female mice were immunized intradermally at the base of the tail with 25 μg type II collagen (induction emulsion medium). Booster immunization with 50 μg of CII in the incomplete adjuvant (IFA) was administered on day 20 after primary immunization. At the end of the study, mice were anesthetized using isoflurane and conducted as humane euthanasia technique, blood was collected by cardiac puncture, and limbs were dissected for histopathology, cytokine quantification, mRNA, and protein quantifications.

### 2.10. Calculation of the Arthritic Score

Arthritis degrees and scores were monitored daily by means of a scale from 0 to 4 for each paw, aiming for a maximum score of 8 per mice. After the induction of arthritis, the joint diameters of the right hind paw were measured using an electronic Vernier caliper (Fischer Scientific, CON3417) as per standard methodology [37], and scoring criteria were followed.

### 2.11. Histopathological Assessment

The hind limb joint tissues were obtained from the mice on day 28 with CIA (*n* = 4; they were selected in a random manner). The tissues were fixed in 10% paraformaldehyde, decalcified in ethylene diamine tetra acetic acid (Sigma-Aldrich, Tokyo, Japan), and embedded in paraffin. The samples were prepared and stained with hematoxylin and eosin (H&E; Merck Millipore, Guyan court, France). H&E-stained sections were used to evaluate the degree of pannus formation (scored from 0 to 4 as follows: 0, normal; 1, minor leukocyte infiltration into the synovium; 2, mild synovium outgrowth; 3, synovium invasion into the joint space; and 4, fibrous ankylosis of the joints). Safranin O-stained sections were used to evaluate the degree of cartilage degeneration (scored from 0 to 4 as follows: 0, normal; 1, minor erosion of the cartilage; 2, mild erosion of the cartilage; 3, partial erosion of the subchondral cartilage; and 4, erosion in all layers of the subchondral cartilage). The average score obtained from the two hind limbs was determined as the score for each mouse, with a maximum possible score of 4.

### 2.12. Immunohistochemistry

Tissue sections were deparaffinized in xylene and then hydrated with gradient alcohol. After antigen retrieval, they were incubated with the corresponding primary antibody overnight at 4 °C. The primary antibody was washed off, and they were they incubated with the secondary antibody for 30 min before being counterstained with Mayer hematoxylin. Two independent researchers observed the positive cells under a microscope [38].

### 2.13. Malondialdehyde (MDA) and Nitric Oxide (NO) Quantification

The expression of oxidative stress and the nitrite levels in limb tissue were estimated. The activity of MPO, a marker of neutrophilic infiltration, and the nitrite levels in µM/mg of tissue were determined according to the method in [39]. Briefly, the limbs were dissected and homogenized using lysate buffer (Invitrogen, Waltham, MA, USA). Cell-free lysate was separated using centrifugation. The absorbance was recorded using a spectrophotometer at 512 nm (Thermo scientific, Waltham, MA, USA). MPO activity was expressed as units per milligram of wet tissue. One unit expresses the MPO activity needed for the conversion of 1 mM of H_2_O_2_ to water in 1 min at room temperature. For NO quantification, the homogenate of the above-mentioned samples was mixed with Griess reagent, and the color intensity of the Griess reagent was modified based on levels of free nitrates in the samples.

### 2.14. Cytokine Estimation

The quantities of cytokines (IL-1β, IL-6, IL-6R, IL-17, and IL-23) were determined in the limb paws of untreated, CIA-, CIA/EAE 10 mg/100 g BW-, and CIA/EAE 20 mg/100 g BW-concentration-treated mice using ELISA kits (Invitrogen, Thermo Fisher Scientific, San Diego, CA, USA and Biolegend Inc., San Diego, CA, USA) following the manufacturers’ protocols. The levels of cytokines in the sandwich-antibody-coated plates were reported at 450 nm on an automated ELISA microplate reader (BioTek Instruments, Winooski, VT, USA). The cytokine values were expressed as pg/mL.

### 2.15. Quantitative Real-Time Polymerase Chain Reaction (qRT-PCR)

Quantitative assessment of the pro-inflammatory and anti-inflammatory mediators and metabolic enzymes in the blood samples of experimental rats was performed by qRT-PCR [38]. Blood was taken from animals and kept in tubes containing EDTA, and the TRIzol process was applied to isolate RNA from blood samples. The purity and final yield were determined by Nanodrop. The cDNA was developed via reverse transcription by applying the protocol of the kit manufacturer. For magnification and quantitative analysis, the standard procedure for the Sybr green master mix (2×) kit was used in qRT-PCR on the Bio-Rad machine. Then, 1 μL of each pair of primers, 7 μL of nuclease-free water, 1 μL of cDNA, and 10 of μLSybr mix were poured into the individual microplate wells. Then, the microplate was shifted to a machine called a thermal cycler set at 45 cycles for denaturation at the temperature of 95 °C, annealing at 60 °C, and termination at 72 °C. The list of primers used in qRT-PCR is shown in Table 1.

### 2.16. Western Blot

Proteins were isolated from limb tissues or joint tissues with RIPA lysis buffer and then separated by 5%–12% SDS–PAGE and blotted onto 0.45 µM of PVDF membranes (Millipore, Bill-erica, MA, USA). These membranes were blocked using 3% skim milk and then exposed to primary antibodies against the following proteins: NF-κΒ (rabbit polyclonal antibody 1:750) (Biorbyt, Cambridge, UK); PTEN (mouse monoclonal antibody 1:1200) (Invitrogen, Waltham, MA, USA); COX-2 (rabbit monoclonal antibody 1:1000) (Biorybt-CB4 0WY, UK-orb621744); STAT-3 (rabbit polyclonal antibody 1:1000) (Biorybt, CB4 0WY, UK, orb126045); OTP (rabbit polyclonal antibody 1:750) (Invitrogen, Waltham, MA USA, PA5-27094); OTC (mouse monoclonal antibody 1:500) (Invitrogen, Waltham, MA, USA, MA3-036); and β-actin (rabbit polyclonal antibody 1:2000) (Cell Signaling Technology, Beverly, MA, USA, 4967S). Then, they were incubated overnight at 4 °C and washed with TBST. Washed blots were incubated with the horseradish-peroxidase-conjugated primary specific secondary antibody at room temperature for 1 h. The blots were visualized by an enhanced chemiluminescence (ECL) system (Pierce, Life Technologies, Austin, TX, USA) and scanned using a LICOR detection system, and expressed bands were analyzed using ImageQuant software and quantified by densitometry using ImageJ software v1.8 [40].

### 2.17. Statistical Analysis

All the outcomes are articulated as the mean ± SEM. The data were analyzed with GraphPad Prism version 5.00. One-way ANOVA followed by Tukey’s post hoc test were used to analyze inflammatory mediators calculated by ELISA, the mRNA gene expression of inflammatory biomarkers, and oxidative stress biomarkers. One-way ANOVA with Student’s *T*-test were applied to assess the paw diameter, scoring, body weight, and hematological and biochemical analysis of CIA-induced arthritis.

## 3. Results

### 3.1. Phytochemical Analysis

The phytochemical analysis of the ethyl acetate and hydro alcoholic extract (70%) obtained from the *OL* cladodes showed the qualitative presence of carbohydrates and/or glycosides, alkaloids, tannins, and flavonoids. Ethyl acetate and hydro alcoholic acid are rich in carbohydrates and/or glycosides, alkaloids, and flavonoids, while hydroethanolic extract contains tannins (Appendix A).

### 3.2. Estimation of Phytochemicals of Opuntia littoralis Cladode Extracts

The results in Appendix A show the total phenolic acid and flavonoid contents in the hexane extract, ethyl acetate extract (EAE), and hydro alcoholic extract (HAE) obtained from *OL*. Significant amounts of phenolic acids and flavonoids were found in the *OL-*EAE and *OL*-HAE. Comparatively, the hexane extract had less phenolic compounds than the other two extracts. Simultaneously, the HAE exhibited the greatest phenolic acid content (88.3 ± 7.1) and flavonoid content (45.2 ± 3.98 μg/mg). In terms of the phenolic acid content, all three extracts showed a >50 mg equivalent to GAE/mg. Plant extracts with >20 μg/mg are considered to have high antioxidant activity. The results showed that the highest phenolic acid content was recorded in the 70% *OL*-HAE, and the most flavonoid-rich extract was *OL*-EAE (Appendix A).

### 3.3. Analysis of Polyphenolic Compounds in EAE and HAE of OL Cladodes Using HPLC

As indicated in Appendix A, the polyphenolic compounds in the EAE and 70% HAE from the *OL* cladodes were detected using HPLC. The *OL-*EAE was found to contain high concentrations of flavonoids, including rutin (3764.54 µg/g), quercetin (2469.22 µg/g), naringenin (2016.78 µg/g), kaempferol (1876.27 µg/g), apigenin (1223.34 µg/g), and hesperetin (984.68 µg/g). In addition, there were also high concentrations of the phenolic acids gallic acid (2912.82, µg/g) and chlorogenic acid (1853.32 µg/g). On the other hand, the *OL*-HAE demonstrated a high concentration of phenolic acids, with gallic acid (4874.53 g/g) and chlorogenic acid (1756.29 g/g) being the two most significant, followed by ellagic acid (304.84 g/g). In addition, three flavonoids were detected: catechin (111.42 µg/g), naringenin (89.81 µg/g), and quercetin (18.71 µg/g).

### 3.4. Effect of OL Extracts on Antioxidant Activity

The reducing power results showed that the EAE and -HAE from *OL* have more potent scavenging activity than the hexane extract when compared with ascorbic acid (Table 2). The hexane extract showed DPPH scavenging activity at 2.22 ± 0.11, and *OL-*EAE and *OL*-HAE showed EC50 concentrations of 1.8 ± 0.12 and 3.2 ± 0.3, respectively. The iron chelating (IC) activity of the tested extracts showed 2.35 ± 0.08, 0.88 ± 0.03, and 1.35 ± 0.02 EC50 for hexane, *OL-*EAE, and *OL*-HAE, respectively. The IC50 values for the ABTS activity of different extracts were 34.41 ± 2.47, 11.99 ± 1.66, and 21.2 ± 1.02, respectively (Table 2). In conclusion, the phytochemical and antioxidant results revealed that *OL*-EAE and *OL*-HAE were selected for further in vitro cellular assays.

### 3.5. Effect of Ethyl Acetate and Hydro Alcoholic Extract on Inflammatory Cells

Synovial stromal cells were stimulated with recombinant LPS antigens, and the synovial cells converted into osteoclastogenesis and reduced the proliferation of stromal cells. The cytotoxicity was estimated to be 40% at the 25 μg/mL concentration treatment. LPS stimulation was active during differentiation and in inflammatory bone cells. *OL*-HAE significantly controlled the differentiation and maturation at the 20 ug/mL concentration (Figure 1A). *OL*-EAE showed less toxicity compared to *OL*-HAE but not many significant changes in toxicity levels. Proliferation was clearly visible via light microscopy, and synovial cells showed a moderate increase in proliferation (Figure 1A). RAW-264.7 cells showed a similar toxicity pattern, and less proliferation was observed compared to synovial cells. The macrophagic cells showed that the pro-inflammatory markers were potently inhibited by both *OL*-EAE and *OL*-HAE, as *OL*-EAE controls the liberation of nitrite in a dose-dependent manner (Figure 1B).

Oxidative stress and lipid peroxidation on LPS-induced inflammatory cells were quantified using malondialdehyde (MDA) liberation. The MDA levels explored in the LPS-treated groups were 0.62 ± 0.021 μM. EAE significantly inhibited MDA liberation at 50 μg/mL, and HAE gradually inhibited the MDA levels in synovial fibroblast cells and was comparatively more significant than macrophagic cells (Figure 1C). The inflammatory cytokines in IL-6-induced synovial cells expressed significant RANKL, IL-23, and IL6R receptors, whereas *OL*-EAE reduced the cytokines as well as chemokines > 25 μg/mL. Macrophagic markers were also reduced, and TNF-α and IL-23 were reduced from 430 pg/mL and 143 pg/mL to 252 pg/mL and 95 pg/mL, respectively (<*p* ± 0.05). *OL-*HAE was comparatively less significant than *OL*-EAE (Figure 1D).

### 3.6. Effect of OL-EAE on Collagen-Induced Arthritic (CIA) Mice

The CIA mice developed severe arthritic progression, swelling, and hind paw deformities. The immunized mice showed symptoms 8 to 28 days after *OL*-EAE 10 mg/100 g BW of mice and 20 mg/100 g BW of mice administration and after 10 days of collagen administration (Figure 2). Arthritic index scores were based on swelling, deformities of limb usage, and erythema and were scored every 7 days. The swelling volume of the injured paws was evaluated, and it was inhibited (*p* < 0.05) after ethyl acetate treatment. At 10 mg and 20 mg treated mice, arthritic index showed decreased values, and significant relief was observed for arthritic symptoms (Figure 2A). Paw edema was significantly increased in the CIA group. It was determined that 20 mg/100g BW of *OL-*EAE reduced the edema level, and significant improvement was observed in the immunized mice after 28 days (Figure 2B). Additionally, the swelling inhibition rate of the 20 mg/100g BW group was similar to that of 10 mg/100 g BW. OL-EAE treated mice did not show significant variation compared to the untreated group (Appendix A).

The ankle flexion pain score was 0 for pain. The ankle flexion of the inflamed paws reached a maximum point on day 1 after collagen injection, and the pain reflex was reduced at the end of the experiment (Figure 2C). The body weights of the mice in the CIA group were lower than those in the ethyl acetate group. However, insignificant body weight changes between both treated groups were observed (*p* < 0.07). The body weights of the mice in the CIA group 14 days after immunization were significantly different compared to the body weights of the group administered *OL-*EAE (Figure 2D). The extract-alone-treated mice showed insignificant variation in paw edema (Appendix A).

### 3.7. Effect of OL-EAE on Macroscopic and Microscopic Assessment of CIA Mice

The anti-arthritic activity of *OL-*EAE in CIA mice and the joint histopathology of the treated mice was assessed. Macroscopic assessments of the collagen-immunized mice showed reduced swelling and redness as well as reduced infiltrations (Figure 3A). The microscopic indices of H&E of CIA mice paws were assessed as cartilage damage and bone erosion in *OL-*EAE in the collagen-immunized mice (Figure 3B). The paws of the CIA mice showed significant degeneration of endothelial tissues and dense connective tissues. Increased cartilage necrosis and fibroblastic cell infiltrations were observed. The hyperplasia and cell exudation in the joint cartilage were significantly attenuated by *OL-*EAE by the 20 mg/100 g BW concentration treatment. A marked reduction in hind paw bone destruction was seen in the *OL-*EAE group (Figure 3B,C). OL-EAE-treated naïve mice showed insignificant variation compared to the untreated group and a pathologically similar pattern to the control mice group (Appendix A).

### 3.8. Effect of OL-EAE on Marker Localization in Arthritic Mice

Cell regulatory markers play a major role in immune pathogenesis in bone deformation. In this study, the therapeutic effect of *OL*-EAE altered the cellular localization of inflammatory mediators. *OL*-EAE treatment significantly decreased NF-kβ, STATE, and PTEN markers in CIA mice, whereas mice that had only been immunized expressed significantly high levels of these markers (Figure 4A,B).

Infiltrated cells producing NF-kβ and STAT-3 in the arthritic joints were observed by immunohistochemical staining and were determined to be moving progressively towards the synovium, especially in synovial endothelial regions (Figure 4A,B). On the other hand, the RANKL marker was significantly sensitized by ethyl acetate treatment and controlled the autoimmune mediators (*p* < 0.01) in CIA model mice (Figure 4). In addition, compared to the *OL*-EAE-treated CIA mice, there were significant increases (*p* < 0.01) in NF-kβ, STATE, and PTEN expression in the CIA-immunized mice.

### 3.9. Regulation of Secondary Inflammatory Mediators in OL-EAE-Treated CIA Mice

MDA and cytokine levels (IL-1β, IL-6R, IL-6, IL-17, and IL-23) were significantly reduced (*p* < 0.01) in *OL*-EAE-treated mice compared to collagen-immunized mice (*p* < 0.05) (Figure 5A). The level of MDA was significantly decreased (*p* < 0.01) compared to the CIA model and in the extract-treated mice. IL-6R is a surface receptor of IL-6 cytokines, and it was gradually balanced based on the level of IL-6. It was cross-checked with in vitro experiments, and similar results to those determined for in vitro IL-6-induced synovial cell behaviors were obtained (Figure 5B). IL-17 and IL-23 are proliferative markers of lymphocytes and regulatory mediators of autoimmune disorders. These two markers were significantly reduced from 69 pg/mL and 192 pg/mL via the *OL*-EAE treatment to 43 pg/mL and 125 pg/mL, respectively (Figure 5B). The extract-alone-treated naïve mice showed significant changes in MDA levels and increased from 0.81 to 1.03 µM compared to the untreated group (Appendix A).

### 3.10. Effect of OL-EAE on Bone Formation and Inflammatory Markers in CIA Mice

The results revealed the inflammatory and osteogenic marker levels in the *OL*-EAE-treated CIA mice (Figure 6A–D). The inflammatory markers of NF-kβ, Stat-3, and PTEN expression were significantly higher in the CIA-immunized mice compared to in the *OL*-EAE-treated mice. mRNA and proteins were sensitive to and had a similar response against the extract treatment. The growth factor VEGF showed increased expression in the extract-treated mice compared to the group of mice immunized with CIA alone.

Furthermore, we investigated more markers related to osteoblastic synovial damage. The correlated mRNA and protein estimations of OTP and OTC were significantly increased in the *OL*-EAE-treated mice compared to in the CIA mice (Figure 6A–D). Increases in protein levels were observed in OTC and OTP in the *OL*-EAE-treated groups. On the other hand, OTP was not significantly upregulated by the extract treatment. Moreover, the group treated with 20 mg of extract showed the significant upregulation of transcript markers of osteoblast-related morphogenic proteins as well as osteo-inductive proteins (Figure 6C–D). Significant observations were noted for the mRNA and proteins of osteo markers.

### 3.11. Effect of OL-EAE Extracts on Epigenetic Regulators (miR28 and miR-199a) of mRNA Targets

Regarding the nuclear receptor interactions and their epigenetic regulators, the impact of the *OL*-EAE extract was evaluated. Through the manipulation of nuclear regulator proteins and COX-2 targeting miR-28 and miR-199a in osteo limb tissue, respectively, the regulation of the epigenetic variables by the *OL*-EAE extract might play a significant role in synovial migration and nuclear breakdown. According to the findings of this investigation, *OL*-EAE therapy significantly upregulated the expression of miR-28 and miR-199 (Figure 7). The virtual bindings of miR28 and miR199a against nuclear protein and COX-2 sequences are supported by computational result data (Figure 7A). Additionally, the reciprocal expressions targeting miR-28 and miR-199a were seen in NF-κΒ and COX-2 (Figure 7B). These modifications showed that *OL*-EAE decreased the levels of pain markers in NF-κB and COX-2 via the modification of the epigenome in host cells.

## 4. Discussion

Rheumatoid arthritis (RA), the most prevalent inflammatory autoimmune disease, is characterized by prolonged inflammation in the synovium tissue and immune cell infiltration, notably **by** macrophages. Treatments made from natural ingredients have been proven to be helpful in meeting the needs of RA therapy [8,20,21]. *O. littoralis* has shown antibacterial and antidiabetic activities in addition to having a high nutritional value [25,26]. Although *Opuntia humifusa* has been shown to be effective in treating and preventing RA, inflammation, and cancer [41], the effectiveness of *OL* in the treatment of RA remains unknown. In the current study, we examined the anti-arthritic properties *OL*-EAE from *OL* cladodes at the concentrations of 10 and 20 mg/g in CIA mice.

In this investigation, the phenolic and flavonoid contents and antioxidant activity of several fractions of *OL* cladode extracts, including hexane, ethyl acetate (EAE), and 70% hydro alcohol (HAE), were quantitatively evaluated. *OL*-EAE showed the highest total phenolic content, followed *OL*-HAE, and the lowest total phenolic content was found in hexane. *OL-*EAE showed high concentrations of flavonoids (rutin, quercetin, naringenin, kaempferol, apigenin, and hesperetin) and phenolic acids (gallic acid and chlorogenic acid).

These findings are consistent with those of Cha et al. [42], who noted that the ethyl acetate fraction of Opuntia humifusa fruit extract stood out for having the highest total phenolic and total flavonoid contents, and that it may be considered to be a reliable source of antioxidants. In addition, our findings concurred with earlier research by Abd El-Moaty et al. [26] and by Mabry et al. [43].

Rutin and quercetin, the two primary components of *OL*, have been shown to assist in pain relief by boosting antioxidant defenses against inflammatory assaults and by lowering the release of inflammatory cytokines [44,45,46]. In addition, derivates of quercetin, isorhamnetin, and kaempferol can influence how cells infiltrate and secrete soluble inflammatory mediators, with important consequences for the inflammatory process [24]. Therefore, we propose that *OL* works synergistically with a number of different components and targets to reduce synovial inflammation and to repair cartilage damage.

In LPS-induced RAW264.7 macrophage cells, *OL*-EAE administration significantly reduces the production of MDA and dose-dependently controls the release of nitrites, demonstrating its antioxidant potential, which may enhance its anti-arthritic effect. These results support earlier research that indicated that *Opuntia humifusa* extract dramatically decreased nitrite generation and iNOS expression in LPS-induced RAW264.7 cells [41,47].

IL-23 mediates chronic joint inflammation via the production of Th17 cells and the induction of TNF-α, IL-1β, and IL-6 in RA [48]. Our data reveal that *OL-*EAE reduced the expression of inflammatory cytokines brought on by LPS, including IL-6R, TNF-α, and IL-23. This suggests that the administration of *OL-*EAE may be able to reduce the release of pro-inflammatory cytokines while preventing the infiltration and destruction of synovial tissue by inflammatory cells.

Based on these prior findings, we assessed the anti-arthritic effects of *OL-*EAE at concentrations of 10 and 20 mg/100 g BW in CIA mice. Acute toxicity tests revealed no adverse effects or fatalities in any of the animal groups, indicating that *OL-*EAE is safe for the treatment of the diseases associated with experimental arthritis.

According to earlier studies, thicker paws are a symptom of arthritis [8,20,21,49]. In the present study, treatment with *OL-*EAE at doses of 10 mg/100 g BW of mice and 20 mg/100 g BW of mice following a 10-day period of collagen immunization led to a progressive improvement in arthritis symptoms. There was a notable decrease in the bone deterioration of the hind paw as well as a considerable reduction in the quantity of paw edemas. This may be explained by the immunological defenses that *OL* extracts may offer, which may halt systemic dissemination and subsequently minimize joint injury in mice.

Our results showed that the arthritic mice lost weight compared to the healthy controls. These results support earlier research indicating that chronic arthritis frequently results in weight loss due to the systemic or local activity of inflammatory cytokines such as TNF-α and IL-1β, which are produced predominantly by the monocytes and macrophages that cause muscle degeneration and loss of appetite [50,51]. Our study demonstrated that administering *OL-*EAE to CIA mice resulted in weight gain. Consequently, plant extracts could be able to reverse the deterioration of the muscles brought on by RA.

NF-κB has been identified as a critical RA marker in the response to inflammation in preliminary research [4]. In addition, IL-6 transmits signals via STAT3 phosphorylation, and STAT3 promotes RA-related joint erosion and inflammation [52]. Furthermore, PTEN expression in RA is regulated by DNA methylation, and PTEN is important in the pathogenesis of RA because it regulates the production and release of pro-inflammatory cytokines and chemokines [16]. The present study found that CIA-immunized mice had considerably elevated levels of the inflammatory markers NF-kB, STAT-3, and PTEN, in line with these suggestions. However, treatment with *OL-*EAE dramatically reduced the levels of the NF-κB, COX-2 STAT3, PTEN, and RANKL markers in the CIA-model mice. Furthermore, the growth factor VEGF showed increased expression in mice treated with extract compared to those immunized with CIA alone. Therefore, our research shows that *OL-*EAE inhibits osteoclast activation and inflammation, acting as a Stat3 inhibitor, and may be a safe therapeutic alternative for RA treatment. Together, the data showed that *OL-*EAE directly contributed to the production of inflammation in mouse models and decreased the infiltration of inflammatory cells. Studies on the expression of miRNAs and epigenetic variables will provide more support.

Furthermore, we found in the present study that IL-1β, IL-6R, IL-6, IL-17, and IL-23, proliferative markers of lymphocytes and regulatory mediators of autoimmune disorders, were also significantly reduced by the *OL-*EAE treatment when compared to the mice in the CIA-control group, suggesting that *OL-*EAE produces an antinociceptive effect in the CIA model. Additionally, the molecular mechanism of the antinociceptive and anti-inflammatory effects of *OL-*EAE may involve the inhibition of pro-inflammatory cytokine levels in the plasma, such as TNF-α, and the suppression of inflammatory mediators such as COX-1, COX-2, NF-κB, TNF-α, and IL-6 in the ankle tissue.

miRNAs are essential for controlling the transcriptome and RA development [12]. Numerous miRNAs influence target genes and pathways, including the NF-κB and JAK-STAT pathways, and are improperly expressed in the cells involved in RA. miR-199a inhibited proliferation and induced apoptosis and can be considered to act as an anti-inflammatory in RA [13,14]. The results of this study show that *OL-*EAE treatment greatly increases the expression of miR-28 and miR-199. Our results indicate that the anti-inflammatory actions of *OL-*EAE altered the cellular localization of the inflammatory mediators, therefore preventing joint inflammation via partial epigenetic and metabolic regulations.

A limitation of this study is that the HPLC data revealed the main flavonoids present in the extract, while more research is needed to understand the genotoxicity, hormonal balance, and use of polyphenols in biosynthesis. Further research is needed to determine the involvement of minor content molecules in arthritis and to assess their function.

## 5. Conclusions

To the best of our knowledge, this is the first study to show that *OL*-EAE protects against bone and cartilage degradation and inflammation during the progression of RA in CIA mice by reducing the release of inflammatory mediators (NF-κB, COX-2 STAT3, PTEN, and RANKL) and pro-inflammatory cytokines (IL-1β, IL-6R, IL-6, IL-17, and IL-23) and by increasing the expression of miR-28 and miR-199. As a result, the findings of this investigation may offer new perspectives on the potential therapeutic use of *O. littorlis* extract for the treatment and prevention of RA.

## Figures and Tables

**Figure 1 nutrients-14-05366-f001:**
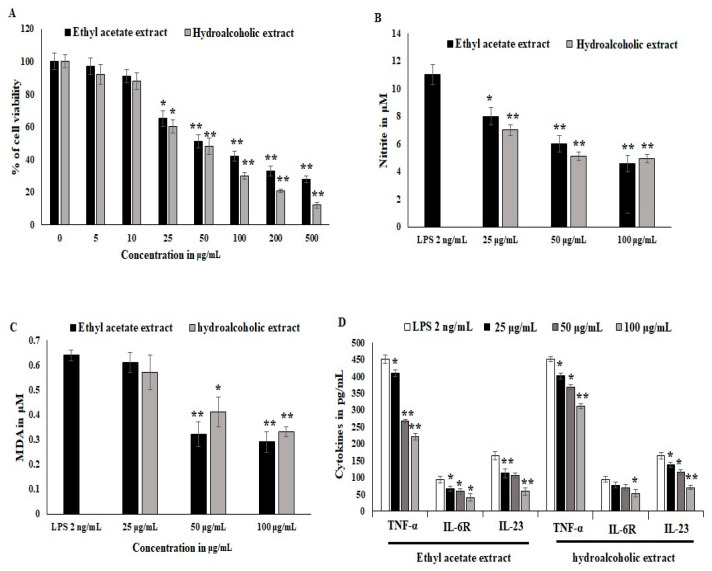
Effects of *Optunia littoralis (OL)* EAE and HAE extracts on cell viability of murine macrophagic cell line RAW 264.7 and mouse synovial-like fibroblastic cells. (**A**) Confluency cell lines were trypsinized and seeded with 10^4^ cells per well in a 96-well plate and set to adhere for 12 h. After adherence, *OL-*EAE and *OL-*HAE extracts were treated at 500, 200, 100, 50, 25, 10, and 5 µg/mL concentrations for 24 h, and the SRB assay was used to determine cell viability after treatment. (**B**,**C**) NO and MDA levels of IL-6-induced RAW264.7 cells were determined by using the Griess reagent method and Malondialdehyde kit (Biovision, Abcam, Boston, MA, USA). Human macrophage RAW264.7 cells were divided into a control group (no LPS was added), LPS group (treated with 2 μg/mL LPS for 24 h), and nontoxic treatment group administered at 10 to 50 µg/mL. NO content in cell supernatant was determined as described in Materials and Methods. MDA values was expressed as µM. (**D**) Inflammatory cytokines TNFα, IL6R and IL23 were quantified using ELISA kit. All the experiments were performed in triplicate as three individual experiments. * *p* < 0.05 and ** *p* < 0.001 for *OL-*EAE and *OL-*HAE extracts and for EAE and HAE extracts with LPS for RAW 264.7 cells.

**Figure 2 nutrients-14-05366-f002:**
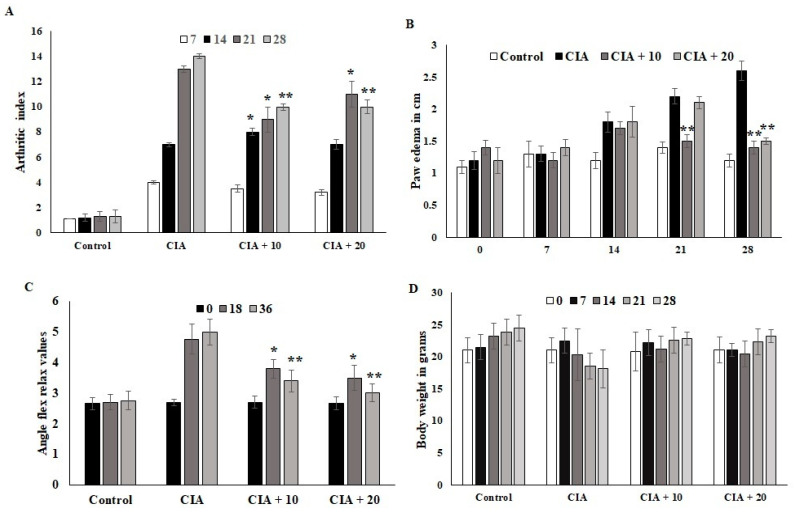
Optunia littoralis (OL) EAE extract suppressed clinical symptoms in collagen-induced mice. Inflammatory arthritic indices of control and CIA-induced mice were evaluated. (**A**) Clinical scoring of joint inflammation was calculated using size in centimeter, redness, hind limb physical movement, lifting and walking, and holding of rods every 10 days. (**B**) Paw edema was calculated according to swelling circumference as determined by Vernier caliper. (**C**) Flex and relax angle was calculated by mobility and climbing activities. (**D**) Body weight was calculated before and after treatment and during every periodic inspection. All the values are expressed as mean ± SEM; each group contained 6 mice; * *p* < 0.05 and ** *p* < 0.001, *OL-*EAE-extract-treated CIA mice compared to disease mice (CIA alone) (*n* = 5).

**Figure 3 nutrients-14-05366-f003:**
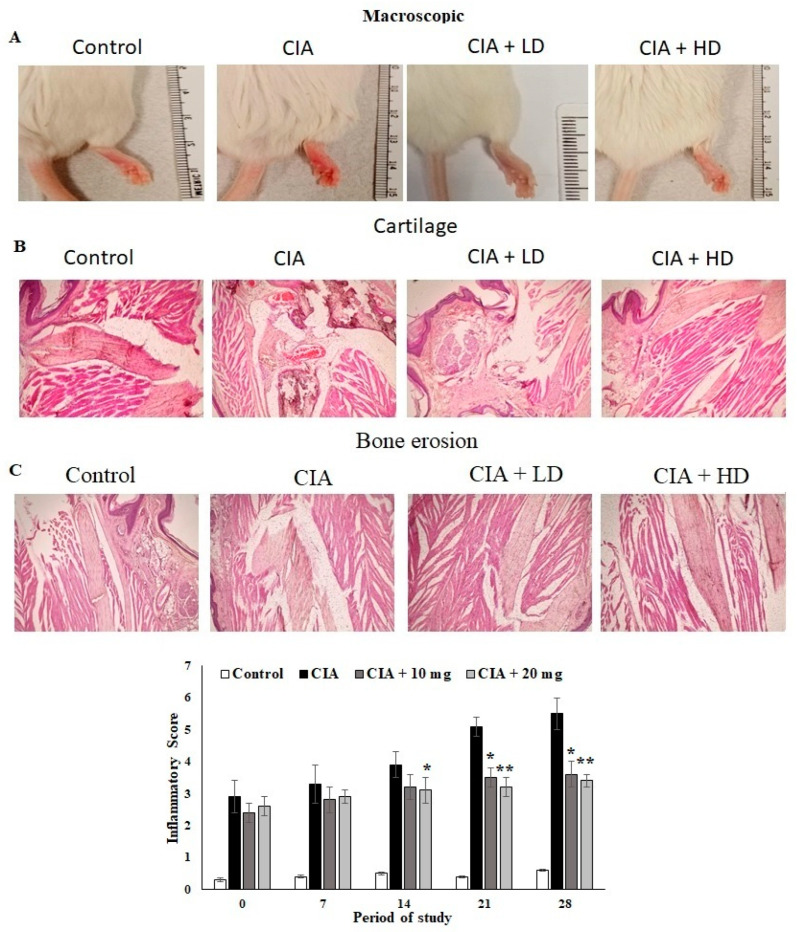
Macroscopic and microscopic examinations of CIA mice treated with EAE extracts. (**A**) Macroscopic observations on the day before the mice were sacrificed on day 27. (**B**,**C**) Histopathological alterations of collagen-induced arthritis in mice. Section of hind limbs of control mice showing normal characteristics. Arthritis mice showing alterations (infiltration of inflammatory cells, INF, destruction of cartilage, CA, bone erosion, BE). PCB-treated CIA mice showing lower number of infiltrated inflammatory cells and cartilaginous cells. Muscle cells with normal appearance were observed. All the values are expressed as mean ± SEM; each group contained 6 mice; * *p* < 0.05 and ** *p* < 0.001 in EAE-extract-treated CIA mice compared to disease mice (CIA alone) (*n* = 6).

**Figure 4 nutrients-14-05366-f004:**
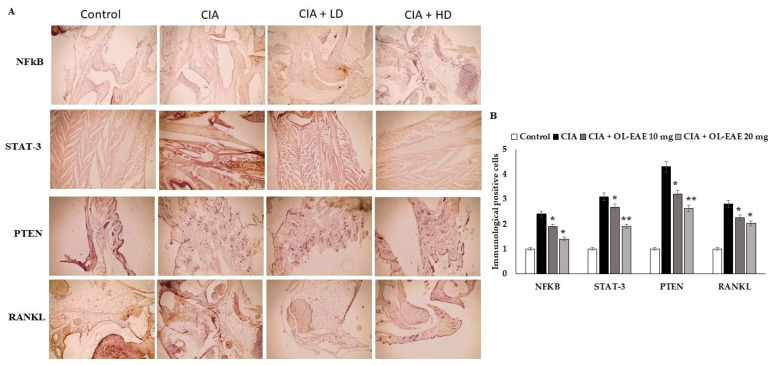
Effects of EAE on NF-kB, STAT-3, PTEN, and RANKL expression in CIA-induced arthritis mice model. (**A**) Representative photomicrographs of immunohistochemical analysis of NF-kB, STAT-3, PTEN, and RANKL in paw sections from control, CIA, low-dose CIA/EAE, and high-dose CIA/EAE groups after 4 weeks. IHC counterstained with Mayer’s hematoxylin, X: 400, bar: 50 µm. (**B**) Histogram plot showing the scores of NF-kB, STAT-3, PTEN, and RANKL represented as fold changes. The data presented are the median (*n* = 6) and * *p* < 0.05 and ** *p* < 0.001 when comparing CIA groups with EAE-treated group using Student’s *T*-test (*n* = 6).

**Figure 5 nutrients-14-05366-f005:**
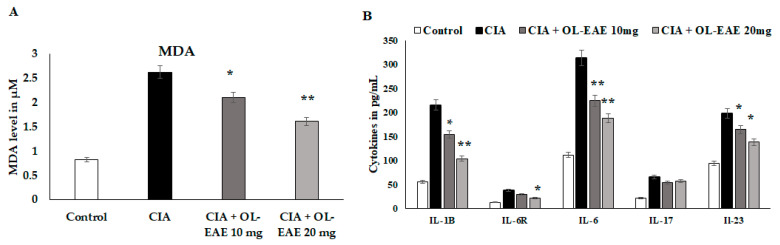
Estimation of oxidative stress marker malondialdehyde and inflammatory cytokines in tissue lysates using ELISA kits and methods. (**A**) The biochemical markers related to inflammation and infiltration were recorded in the CIA mice. The MDA level was expressed in µM/mg of protein. (**B**) IL-1β, IL-6R, IL-6, IL-17, and IL-23 in the tissue of vehicle-control mice and CIA mice after receiving repeated concentrations of *OL*-EAE 10 and 20 mg/100 g mouse weight for 28 days. Samples from groups comprising five mice were analyzed. Quantitative data (mean ± SD) are presented using the average optical density values at 450 nm in a microplate reader; * *p* < 0.05 and ** *p* < 0.001 significance was used to compare CIA mice to CIA/*OL*-EAE-treated mice (*n* = 6).

**Figure 6 nutrients-14-05366-f006:**
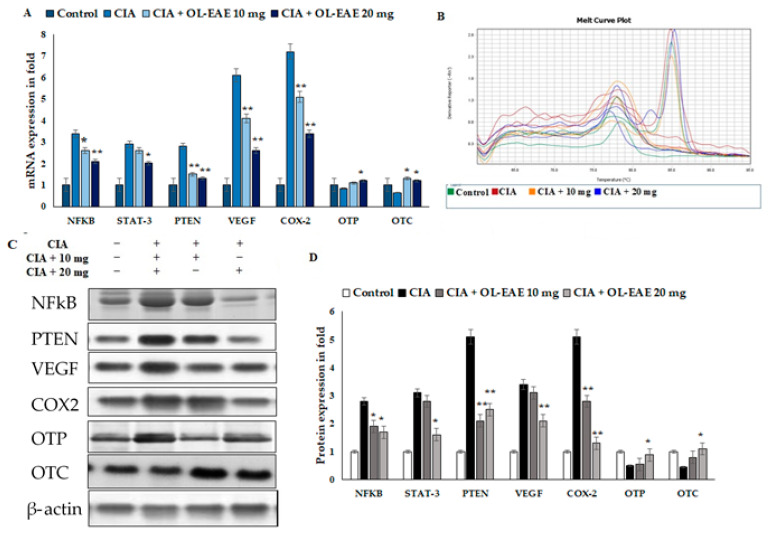
Effects of *OL*-EAE extract on NF-κB, STAT-3, PTEN, VEGF, COX-2, OTP, and OTC signaling. Metabolic markers were quantified using real-time PCR and Western blot methods. (**A**) The inhibitory effects of *OL*-EAE extract after 28 days of treatment with 10 and 20 mg extract concentrations suppressed the inflammatory mediators in both mRNA and protein levels. Quantification was determined via RNA extraction, cDNA preparation, and the quantification of specific genes using real-time PCR. The gene expression levels were quantified using the ΔΔcT method. (**B**) Melting curve of PCR reaction determined the calibrated gene expression of this experiments (**C**,**D**) The cell lysate was collected and subjected to Western blot analysis. The primary antibody of each target was from the monoclonal antibodies from abbexa, abcam, and biorybt. The density of each band was analyzed using ImageJ v1.6. Each value is expressed in triplicate as the mean ± SD; * *p* < 0.05 and ** *p* < 0.001 when comparing the CIA disease group with the CIA/*OL*-EAE-treated group (*n* = 6).

**Figure 7 nutrients-14-05366-f007:**
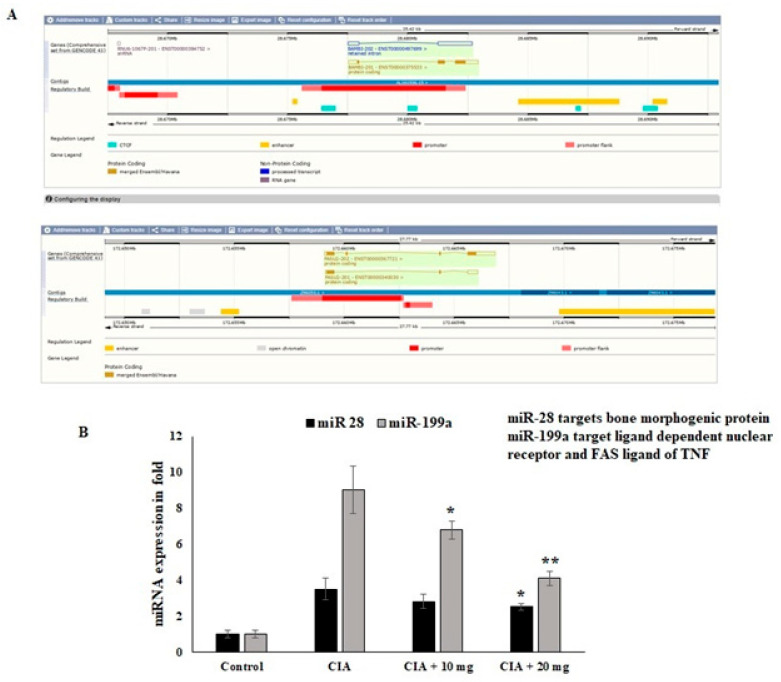
Effects of *OL*-EAE extract on mRNA-targeted miRNA expression levels in paw limb tissues: miR-28 and miR199a are the targets of NF-kB and COX-2. (**A**) The miRNAs abrogate the autoimmune inflammatory conditions in osteo sites. (**B**) The expression levels of mRNA coding targets such as miR-28 and miR-199a were compared with U6 internal control CIA, low-dose CIA/*OL-*EAE, and high-dose CIA/*OL-*EAE groups. Each value is expressed in triplicate as the mean ± SD; * *p* < 0.05 and ** *p* < 0.001 when comparing the CIA disease group with the CIA *OL-*EAE-treated group (*n* = 5).

**Table 1 nutrients-14-05366-t001:** Primers used in qRT-PCR.

Primer Name	Forward Sequence	Reverse Sequence	PCR Product Size
NF-KΒ	CATGAAGAGAAGACACTGACCATGGAAA	TGGATAGAGGCTAAGTGT AGACACG	198
COX-2	TGTATGCTACCATCTGGCTTCGG	GTTTGGAACAGTCGCTCGTCATC	147
PTEN	TGTGGTCTGCCAGCTAAAGG	ACACACAGGTAACGGCTGAG	192
VEGF	TGCAGATTATGCGGATCAAACC	TGCATTCACATTTGTTGTGCTGTAG	187
OTP	ACATCCAGTACCCTGATGCTACAG	TGGCCTTGTATGCACCATTC	204
OTC	AGCAAAGGTGCAGCCTTTGT	GCGCCTGGGTCTCTTCACT	176
GAPDH	GCAAGGATACTGAGAGCAAGAG	GGATGGAATTGTGAGGGAGATG	113

**Table 2 nutrients-14-05366-t002:** Antioxidant activity of all *Opuntia littoralis* extracts.

Plant Extracts	EC50 Values (μg/mL) of Radical Scavenging		
	Hexane Extract	*OL-*EAE	*OL-*HAE
DPPH	2.22 ± 0.11 a	1.8 ± 0.12 a	3.2 ± 0.3 b
FRAP	2.35 ± 0.08 a	0.88 ± 0.03 bc	1.35 ± 0.02 b
ABTS	34.41 ± 2.47 a	11.99 ± 1.66 c	21.2 ± 1.02 bc

Each value in the table is represented as mean ± SD (*n* = 3). Values in the same column followed by a different letter (a–c) are significantly different (*p* < 0.05).

## Data Availability

Not applicable.

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
