# Peer review of "Phenolic-Compound-Rich Opuntia littoralis Ethyl Acetate Extract Relaxes Arthritic Symptoms in Collagen-Induced Mice Model via Bone Morphogenic Markers"

_nutrients, 2022, doi:10.3390/nu14245366_

Round 1

Reviewer 1 Report

1. Specific number of mice should be included in the figure legends. 

2. The discussion should be re-written and shorten. The authors didn't need to list all results one by one, and can show some instructive results. 

3. Limitation should be included in the discussion. 

Reviewer 2 Report

The authors conducted a study trying to investigate the anti-inflammatory effect of Opuntia littoralis in collagen-induced arthritis mouse model. The authors ran a lot of assays, including chemical analysis of the plant extract to measure the polyphenol amounts, an in vitro assay to assess the anti-inflamatory effect of the extract, and an in vivo assay to understand if this plant extract can serve as potential preventive or therapeutic agents against Rheumatoid arthritis. Despite the huge workload, I think this manuscript has some major experimental design issues, especially in the in vivo study. The writing also does not meet publication quality. It needs substantial structure change and extensive language review. The typos, inconsistency, and missing information make reading the manuscript very hard. I recommend the authors reading more publications on the similar topic to learn how to design and execute a proper experiment using the CIA model, as well as how to put together the results in a manuscript. An example is here: https://www.ncbi.nlm.nih.gov/pmc/articles/PMC16365/

Manuscript structure suggestions:

Please use supporting information. The authors did many, if not repetitive, chemical analyses to identify the polyphenol components in the plant extract. This part is not the biggest highlight in this study. Please shorten the related description, Table 2-4, and Figure 1 and only describe the result of bioactive component category (polyphenol specifically since they are likely related to the following bioactivities) and their concentrations. Additional information can be provided in the supporting information.  

Please give some description on CIA mouse model in the Introduction.

Method: in addition to the aforesaid issue, details of chemicals, reagents, antibodies used in the immunohistochemistry, etc. are needed.

Discussion: there are already many studies on understanding the effect of other polyphenols in the same animal model. The authors can make comparison to these previous studies. Does Opuntia littoralis make a better therapeutic agent compared to other plant extracts?

Experimental design questions:

Lines 157-161: section 2.6.

·       How was the administration performed? Check the language. Describe it in line 159. The administration of the extracts in the in vivo experiment is also not clear. I can only guess they are both oral.

Lines 162-173: section 2.7. Too much information lacking here and too much typo and language issues. It is very hard to understand how the animal experiment was conducted.

·       The current experimental design lack a control group of mice receiving only OL-EAE. The naming of groups here are not consistent with the rest of the manuscript. The authors used at least four different ways to name their groups. Please keep the naming consistent.

·       Swiss albino mice: please give genetic background of the mice strain. The mice used in the study should be susceptible to collagen-induced arthritis

·       Collagen: how was this solution prepared? How was the collagen immunization performed? By daily i.v. injection or one-time injection? Why i.v. administration rather than intradermal administration? Why not following the widely used CIA protocol (https://link.springer.com/protocol/10.1385/1-59259-771-8:207)? Would i.v. induces more immunal response compared to intradermal administration? What is the dosage of collagen? Please check your units. They look very strange.

·       How was the OL-EAE administered? Through oral gavage? What is the i.p. administration of saline for in control? Is it a typo for i.v. since collagen was administered through i.v.?

Figure 5-7. When comparing cytokine/protein/mRNA across different treatment groups, please use the cytokine/protein/mRNA as the x-axis and label the bars with treatments. The current way makes interpreting the effect of different treatment very hard. Also I don’t understand the statistical significance indicate which pair of post-hoc comparison.

Round 2

Reviewer 1 Report

The authors well revised the manuscript according to the comments. 

Author Response

Thank you very much. Authors appreciate the reviewer interest to improve the manuscript

Reviewer 2 Report

Thank you for revising the manuscript and providing the replies. The manuscript looks better. However, some of my questions are still not properly addressed. Unfortunately, I think these questions are very important and I was hoping for better solutions. My questions are still around the in vivo study design and its description (2.8. Experimental animals)

  • I still don’t understand why the author described the collagen dosage as “25 μg/25 mg”. Given that I have asked the authors to double check the units in this section, I assume the same numbers and units in the revised manuscript indicate the actual dosage the authors used in the study, which makes the treatment very confusing. From the newly added ”Briefly, female mice were immunized intradermally at the base of the tail with 25 μg type II collagen (Molequle On, New Zealand) in 4 mg/mL of complete Freund’’s adjuvant (CFA)”, it seems that one mouse received 25 μg of collagen regardless of body weight. This roughly equals to 25 μg/25 g, which is very different to “25 μg/25 mg”. What exactly was the dosage in the immunization?
  • Similarly, the dosage for OL-EAE is “10 mg/100 mg” or “20 mg/100 mg”. If these are true, each mouse was gavaged with approximately 2.5 g extract each time. This is a surprisingly high amount.
  • My comment “The current experimental design lacks a control group of mice receiving only OL-EAE” was not addressed. This animal study has two factors (CII and OL-EAE), and having their own controls for these two factors would make this study more scientifically sound.
  • In the future, please put the collagen preparation into an independent section (potentially titled “Preparation of type II collagen emulsion for immunization”) and put it before the immunization and treatment. Change the title “experimental animals” to “Immunization and treatment” or something similar. “Experimental animals” usually only describes the information of the animal used in the study, like genetic background, gender, living situation, etc.
